# Satisfaction with Life and Nutritional Behaviour, Body Composition, and Functional Fitness of Women from the Kraków Population Participating in the “Healthy Active Senior” Programme

**DOI:** 10.3390/ijerph20031877

**Published:** 2023-01-19

**Authors:** Maria Gacek, Agnieszka Wojtowicz, Grażyna Kosiba, Magdalena Majer, Joanna Gradek, Agnieszka Koteja, Olga Czerwińska-Ledwig

**Affiliations:** 1Institute of Biomedical Sciences, University of Physical Education, 31-571 Krakow, Poland; 2Institute of Social Sciences, University of Physical Education, 31-571 Krakow, Poland; 3Institute of Sports Sciences, University of Physical Education, 31-571 Krakow, Poland; 4Institute of Basic Sciences, University of Physical Education, 31-571 Krakow, Poland

**Keywords:** senior women, satisfaction with life, nutritional behaviours, somatic indices, functional fitness

## Abstract

Pro-health behaviours are related to a person’s personal resources. The aim of the study was to assess the relationship between satisfaction with life (SWL), nutritional behaviours, somatic indices, and functional efficiency of senior women. The research was conducted among 120 women aged 60–84 (Me = 65) participating in the “Healthy Active Senior” project at the University of Physical Education in Kraków. The Satisfaction with Life Scale (SWLS) and the proprietary validated questionnaire of nutritional behaviour were used. Body composition was assessed using the method of bioelectrical impedance (TANITA SC-330ST analyser), while physical fitness was evaluated via the Senior Fitness test (Fullerton Functional Fitness Test). Correlations between the variables were measured by implementing Spearman’s R signed-rank correlation coefficients (with *p* < 0.05). Positive correlations between SWL and selected nutrition behaviours have been demonstrated, including eating 5–6 meals (*p* < 0.001) and drinking at least 2 litres of fluids a day (*p* = 0.023), consuming cereal products daily, including whole-grains (*p* = 0.001), avoiding alcoholic beverages (*p* = 0.030), and applying vitamin D supplementation (*p* = 0.010). At the same time, negative correlations between SWL and limiting the consumption of red as well as processed meats (*p* = 0.002), animal fats (*p* = 0.046), and the preference for vegetable oils in one’s diet (*p* = 0.023) were shown. Significant correlations between satisfaction with life and two indicators of functional fitness were also confirmed: negative—with the variable ‘2.44-m Get-Up and Go’ (*p* = 0.003); and positive—with the ‘2-Minute Step in Place’ test (*p* = 0.034). The relationships between SWL and somatic indices did not reach the level of statistical significance. Among the women participating in the “Healthy Active Senior” programme, correlations between SWL and rational nutritional behaviours, as well as indices of functional fitness, were found (mostly positive), while the trends in these areas were not fully unambiguous, suggesting the validity of conducting further research.

## 1. Introduction

A key factor in improving health potential, preventing chronic diseases, and delaying involutional changes is maintaining a healthy lifestyle, including a rational model of nutrition and recreational physical activity [1,2,3]. Health potential is positively influenced by a varied and balanced diet, rich in products with high nutritional density (including vegetables and fruits, whole-grain cereals, fermented milk products, fish, nuts), while limiting the consumption of high-energy density products, rich in saturated fatty acids, cholesterol, trans isomers of unsaturated fatty acids, and simple sugars [4,5,6,7], including the Mediterranean diet [8]. An element of maintaining health is also health training, which includes not only endurance and resistance exercises, but also balance, mobility, and agility interventions. These positively affect, among others, physical fitness and somatic indices [9,10,11,12]. In seniors, this is even more important as aging is related to, inter alia, changes in body composition and a decrease in functional efficiency, understood as the ability to safely and effectively perform everyday activities. Such changes further reduce quality of life [13,14,15,16,17,18,19]. Despite the key importance of behavioural factors for health and quality of life, in numerous studies, nutrition-related mistakes have been indicated, as well as a low level of physical activity among seniors, both in Poland [20,21,22] and in other countries [23,24,25,26].

Pro-health behaviours, including nutritional ones, are determined by numerous factors, i.e., those environmental and individual [27]. Among the psychological features significant for shaping health culture, personal resources occupy an important place, including sense of satisfaction with life (SWL), which is a subjective measure of well-being related to the cognitive assessment of life quality [28,29]. In the research, the predictive importance of personal resources was confirmed, including SWL, for the quality of food choices in various population groups, including perimenopausal women [20,21] and females with type 2 diabetes [30]. In the study, significantly different forms of physical activity for the improvement of SWL in senior women were observed [31,32,33,34,35,36]. On the other hand, in Finnish studies, the importance of physical activity and the Baltic Sea diet have been confirmed to influence SWL among senior women, with the authors additionally paying attention to the small number of studies on the correlations between diet, physical activity, and satisfaction with life [37]. The significance of satisfaction with one’s personal life for health was also noted in the research, as women who are more satisfied with their lives demonstrate lower body mass, better visual–auditory coordination and flexibility of the spine as well as lower blood cholesterol levels [38] and higher bone mineral density [39]. On the other hand, reduced SWL is associated with a higher risk of chronic diseases, especially in women. This has been confirmed in prospective German studies [40]. It may be summarised that diet is significantly related to well-being and health [41].

The “Healthy Active Senior” project implemented at the University of Physical Education in Kraków from 2019, aimed at improving the psychophysical fitness of seniors by participating in recreational activities and lectures on modified conditions of the aging process and opportunities, fits in with the postulate of promoting a healthy lifestyle and delaying involutional changes. Nordic walking, Smovey^®^, BodyArt, and aqua aerobics classes (for 9 months, lasting 90 min, twice a week, 3 selected modules for 3 months) were part of the health training intervention. During the implementation of the “Healthy Active Senior” programme, research was conducted in order to assess SWL, nutritional behaviours, somatic and functional fitness indices, as well as correlations between the intensity of satisfaction with life and nutritional behaviours, the level of somatic indices and indicators of functional fitness among women. This research is included in studies on assessing the determinants of holistically defined health.

## 2. Materials and Methods

### 2.1. Participants

Research was carried out in the fall of 2019 among women participating in the “Healthy Active Senior” project, implemented at the University of Physical Education in Kraków (POWR.03.01.00-00-T225/18) under the EU Operational Programme—Knowledge, Education, Development for 2014–2020. Information about the planned programme was posted on the University of Physical Education website and disseminated by the representative of the Mayor of the City of Kraków for the elderly, e.g., through Krakow’s centres of daily activity for seniors and the local press. Access to the programme was open to people aged 60+ (until the limit of places, set at 136 people, was exhausted). The study was conducted after obtaining the participants’ written informed consent, which was done in accordance with the principles of the 1964 Declaration of Helsinki.

The study included 120 women aged 60–84 (M = 65.56; SD = 4.15, Me = 65). The sample was dominated by women with higher (58.33%) and secondary education (37.50%), there were fewer women with a basic vocational level of education (4.17%). The majority of the women lived in Kraków (91.66%), less frequently in smaller centres. They stated their financial situation to be average (75.00%), less often as below average (12.50%) and above average (12.50%). In terms of health evaluation, women declared the presence of chronic diseases, most often hypertension (33.33%), type 2 diabetes (12.50%), and osteoporosis (12.50%).

### 2.2. Instruments

Satisfaction with life was assessed using the Satisfaction with Life Scale (SWLS) [42], in the Polish adaptation by Juczyński [28]. The results of the SWLS, consisting of 5 items, range from 5 to 35 points, while the higher the score, the higher the level of satisfaction with life.

To assess nutritional behaviours, the author’s (M. Gacek) questionnaire for assessing eating behaviours among seniors was used, referring to the Polish recommendations for this age group [43]. The questionnaire consists of 22 items regarding the degree of implementing rational eating behaviours, using the 5-point Likert response scale (from 1 to 5, denoting “definitely not”, “rather not”, “hard to say”, “rather yes”, to “definitely yes”). The questionnaire concerned: eating the recommended number of meals (5–6 and 4–5), regularity of meals, eating a hot meal during the day, drinking at least 2 litres of fluids, consuming recommended products (vegetables and fruits, wholemeal cereal products, dairy products, legumes and other protein products), preferred fats, limiting the consumption of non-recommended products (red meat and meat products, sugar, sweets and sweetened beverages, salt, alcoholic beverages), the use of spices, vitamin D supplementation, and moreover, individualisation of diet depending on state of health, taking food and drug interactions into account, and level of undertaken physical activity. Based on the results of the questionnaire, the degree of implementing individual dietary recommendations and the general index of rational eating behaviours were assessed (on a scale of 1–110 points, assuming that the higher the index, the more intense the rational eating behaviours). Cronbach’s α coefficient for the author’s questionnaire under discussion was 0.74, which proves that the tool is of sufficient internal consistency. Sufficient validity of the tool was also confirmed by correlation with the pro-healthy diet index (pHDI-10) (R = 0.25; *p* = 0.018). The health-promoting diet index is associated with the consumption of products with potentially positive impact on health (wholegrain bread, other wholegrain cereal products, milk, fermented dairy drinks, cottage cheese, white meat, fish, legumes, fruits, and vegetables) [44].

Body composition was also assessed via electrical bioimpedance using the TANITA SC-330ST body composition analyser according to standard methodology. Within this range, body mass, (kg), body fat (% and kg), lean mass (kg), and total water content (kg) were evaluated.

Physical fitness was examined using the Senior Fitness test (Fullerton Functional Fitness Test) [13]. The test consists of six fitness tests that allow indirect assessment of upper and lower body strength, flexibility, complex motor coordination and balance, and aerobic endurance. Subsequent tests of the Fullerton test include: (1) flexing the loaded arm for 30 s (assessment of upper body strength—arm muscles, ‘Arm Curl’), (2) back scratch, reaching behind one’s back, so-called ‘Safety Pin’ test (assessment of mobility, flexibility of the upper body—upper limb, ‘Back Scratch’), (3) getting up from a chair within 30 s (assessment of the strength of the lower body—pelvic muscles and lower limb, ‘30-Second Chair Stand’), (4) sitting and reaching, bending the trunk forward in a seated position (evaluation of agility, flexibility of the lower body), (5) stand up and walk 2.44 m (assessment of agility and dynamic balance, Timed Up and Go), (6) 2-min walk in place (2-Minute Step, endurance and exercise tolerance assessment) [13].

### 2.3. Statistical Analyses

Descriptive statistics (M, SD, Me, Min, Max) were calculated using the IBM SPSS 21 statistical package. Due to the nature of the distribution regarding variables, the median was applied as a measure of the central tendency. The correlations between the analysed variables were assessed with the use of Spearman’s R signed rank correlation coefficients. Regression analysis was also performed, assuming a significance level of *p* < 0.05.

## 3. Results

In the group of women under study, the median level of satisfaction with life (on the SWLS) was 21 (M = 22.42, SD = 4.67, Min = 9.00, Max = 32.0).

Among the assessed rational nutrition behaviours, to the highest degree (Me = 5, which is ‘definitely yes’), the examined women consumed at least 1 hot meal a day, vegetable fats every day (or almost every day), used spices, took vitamin D supplements, and adjusted their diet to their health condition. To a lesser extent (Me = 4, i.e., ‘rather yes’), they usually ate 4–5 meals a day, consumed varied meals, drank at least 2 litres of fluids daily, consumed several (3–5) portions of fruit and vegetables per day, as well as cereal products, including whole grains, fish, eggs, meat and lean meat, legumes, while limiting the consumption of red and processed meats, as well as animal fats, sugar, sweets and sweetened drinks, salt, and alcoholic beverages, while also paying attention to interactions between foods and medications, and undertook recreational physical activity for about 30–45 min every day. To a lesser extent, they consumed dairy products, including fermented ones, and sea fish at least once a week (Me = 3, which is, ‘difficult to say’). At the level of: ‘rather not’, they consumed 5–6 meals a day (Me = 2.5). The median of the overall index regarding rational food choices was 87 (on a scale of 1–110 points) (Table 1).

In the studied group of women, the median for BMI was 26.65 kg/m^2^, adipose tissue accounted for 33.68% of body mass, the amount of lean mass totalled 44.55 kg, and the water content equalled 32.60 kg (Table 2).

Among the analysed indices of functional fitness (for the Fullerton test), the median results in the ‘Stand-Up from Chair’ within 30 s was 15 repetitions, in the ‘Flexing Loaded Arm’ within 30 s was 18 repetitions, in the ‘Seated Forward Bend’ test, women obtained a result of 5 cm, for the so-called ‘Safety Pin’ (Back Scratch) test—the result of 1 cm was obtained, in the ‘2.44-m Get-Up and Go’ test, the result was 4.67 s, and in for the ‘2-Minute Step in Place’—the result of 111 leg lifts (steps) was noted (Table 2).

Statistical analysis allowed to demonstrate positive correlations between SWL and certain nutrition behaviours, including the consumption of 5–6 meals a day (*p* < 0.001), drinking at least 2 litres of fluids per day (*p* = 0.023), daily consumption of cereal products, including whole-grains (*p* = 0.001), avoiding alcoholic beverages (*p* = 0.030), and daily supplementation with vitamin D (*p* = 0.010). At the same time, negative correlations were noted between SWL and limiting the consumption of red as well as processed meats (*p* = 0.002) and animal fats (*p* = 0.046), and a preference for using vegetable oils in one’s diet (*p* = 0.023). Relationships between satisfaction with life and other eating behaviours (including, among others, the consumption of fruit and vegetables, dairy products and fish, and limiting sweets and salt in one’s diet) did not reach the level of statistical significance (Table 3).

Statistical analysis allowed to confirm significant correlations between SWL and 2 indicators of functional fitness: negative with the variable: ‘2.44-m Get-Up and Go’ test (*p* = 0.003), and positive with the ‘2-Minute Step in Place’ attempt (*p* = 0.034). Positive correlation between life satisfaction and the results of flexing the loaded arm within 30 s were also close to the assumed level of significance (*p* = 0.050).The correlations between satisfaction with life and remaining indices of functional fitness (getting up from a chair withing 30 s, forward bow in seated position and back scratch) and with somatic indicators (body mass, BMI, fat, fat-free mass, and water content) did not reach the level of statistical significance (Table 4).

Regression analysis was also performed to find out which of the variables could explain the level of SWL in the studied group of senior women (Table 5). The following variables were used in the model: age, the scale of rational eating behaviours as well as indicators of functional fitness and somatic indices. The model explains 28% of the variance in SWL (*p* < 0.001). The following predictors turned out to be statistically significant: age, the scale of rational eating behaviours (positive correlation) and the functional fitness index ‘2.44-m Get-Up and Go’ (negative correlation). The influence of other analysed variables did not reach the level of statistical significance.

## 4. Discussion

In the research under discussion, statistically significant correlations have been demonstrated between the level of satisfaction with life and selected nutritional behaviours as well as functional fitness indicators. However, no significant correlations were found between SWL and the analysed body composition indices noted among women from the Kraków community participating in the “Healthy Active Senior” programme at the initial stage of its implementation.

In turn, when discussing the analysed variables, it should be indicated that the studied women participating in the programme to promote health among seniors, an average level of SWL was shown (Me on the SWLS = 21).

In other Polish studies, similar levels of SWL have been demonstrated, e.g., in a group of perimenopausal women (M = 20.4, SD = 5.5), and among women with type 2 diabetes (M = 20.8, SD = 5.3) [21,30]. Therefore, it may be concluded that the level of satisfaction with life among the senior women under study did not significantly differ from the values described in other groups of women, both those healthy and suffering from chronic diseases.

Moreover, in our research, women declared an average level of rational eating behaviours, implementing various qualitative nutritional recommendations to a varied degree (index of rational eating behaviours—Me = 87/110 points). The most common rational food choices concerned: eating at least 1 hot meal daily, consuming vegetable fats every day, using spices, vitamin D supplementation, and adjusting one’s diet to health state. On the other hand, to a limited extent, the women followed the recommendations regarding the consumption of dairy products and sea fish as well as consuming 5–6 meals a day. The described nutritional mistakes could increase the risk of calcium and omega 3 PUFA deficiencies, components of significance in the prevention of osteoporosis (Ca) and cardiometabolic diseases (Omega 3 PUFA), which could be related to the chronic diseases declared by the studied women (described in the ‘Methods’ section). The authors of many publications indicate the pro-health nature of a diet rich in fruits, vegetables, milk and dairy products, legumes, and oilseeds, as opposed to processed foods, sweets, products with a high fat content and alcohol [5,6,45,46].

In Finnish research, the importance of healthy Baltic Sea and Mediterranean diets for the bone mineral density of senior women has been indicated. Higher bone density scores were associated with greater consumption of fruits and berries, vegetables, fish, and low-fat dairy products, and a limited consumption of processed meats [47]. Low health quality of one’s diet, including low protein intake, is a risk factor for the health of older individuals, including frailty and sarcopenia [48,49,50,51,52,53].

Referring to the results of our research to those obtained in other studies, low health quality of the diet and nutritional disorders have been noted among women of all ages, including senior women [20,21,26,44,54,55,56].

In terms of somatic indices, our research allowed to indicate that having a median BMI (26.65 kg/m^2^) is considered as being overweight (according to WHO). The achieved values of the body composition indices (e.g., Me for the amount of adipose tissue equal to 33.68%) suggests slightly increased fat content in body mass, in relation to the recommendation that 33% fat content is a critical value, increasing health risks [57]. In other studies, different levels of somatic indicators have been shown, including the values of BMI and body composition similar to those achieved in our research [58,59].

The assessment of functional fitness among the studied women based on the Fullerton test showed low and average values in various aspects, similar or lower than those described in other studies conducted by various authors in other groups of physically active senior women in [58,59].

Assessment of correlations between the level of SWL and the nutritional behaviour of women participating in the “Healthy Active Senior” programme demonstrated ambiguous tendencies, because on the one hand, positive correlations were obtained between SWL and more rational food choices (in terms of consuming the recommended number of meals a day, an adequate amount of fluid intake, consumption of whole-grain cereal products, avoiding alcoholic beverages, and supplementation with vitamin D), while on the other, negative correlations were noted between SWL and rational food choices (in terms of reducing the consumption of red and processed meat products as well as animal fats and preferring vegetable oils). Therefore, high satisfaction with life was conducive to behaviours having positive impact on the rate of metabolic processes, proper fluid replenishment, securing the supply of complex carbohydrates (the basic source of energy), dietary fibre (with numerous functional properties) and vitamin D (with pleiotropic properties, including those important in prevention of osteopenia and osteoporosis). At the same time, however, this was also associated with behaviours increasing the risk of disturbed fatty acid profile in one’s diet (saturated vs. unsaturated), which may indirectly indicate a lack of health problems requiring such correction of diet in people with high SWL. Nonetheless, it should be noted that the number of positive food choices, along with an increase in the level of SWL was greater than the number of negative choices. Similar tendencies towards making rational food choices among people with a higher level of satisfaction with life (and the sense of disposable optimism correlated with it, were also noted in other studies among various population groups, including perimenopausal women [20,21] and females with type 2 diabetes [30].

Correlations between quality of diet and the physical as well as psychological domain of the quality of life among Australian elderly women have also been demonstrated [60]. Research carried out in a group of senior women from rural areas in Japan also allowed to find a relationship between satisfaction with the quality of nutrition and satisfaction with life [61]. In the research, it was also confirmed satisfaction with one’s personal life also has an influence on health. It was found that women who were more satisfied with their lives had lower levels of body mass and blood cholesterol [38], as well as higher bone mineral density [39], which could be related, inter alia, to a more rational model of nutrition.

Moreover, assessment of the relationship between the level of SWL and indicators of functional fitness among women participating in the “Healthy Active Senior” programme showed ambiguous trends. On the one hand, positive correlations were demonstrated between SWL and higher results of the ‘2-Minute Step in Place’ test, while on the other, a negative relationship was noted for the variable—“Get-Up and Go 2.44 m’, which means that higher SWL was positively correlated with exercise tolerance, endurance, and physical performance, but negatively with agility and dynamic balance. Due to the fact that the level of endurance is associated with undertaking regular aerobic physical activity, it can be concluded that women performing health training show higher SWL, which would indicate a significant level of physical activity for holistically defined health. This was confirmed in other studies, proving that women between the age of 50 and 55 who regularly engaged in recreational physical activity at fitness clubs more often declared SWL than less physically active women [62]. Similarly, in Chinese research, a significant correlation was indicated between positive associations of moderate and high physical activity and the level of life satisfaction among young, middle-aged, and elderly people [35]. In other studies, the positive effect of Guozhuang Dance activity has been demonstrated on the well-being of seniors, including the chain mediating effect of group identity and self-efficacy [36]. In American research, it was also shown that limited physical fitness was a significant predictor of lower SWL in senior people [63].

However, in our research, no significant relationships were noted between satisfaction with life and the analysed somatic indices (body composition). On the other hand, in other studies, correlations have been demonstrated between somatic indicators of nutritional status and the quality of life among senior women. In this respect, it was shown that the women’s quality of life decreased along with an increase in body mass and WHR [64,65]. Moreover, among women with osteoporosis, it has been shown that the highest level of satisfaction with life was declared by women with normal body mass [66]. In turn, in other studies, it was indicated that women with excessive body mass declared lower satisfaction with life [67].

The obtained regularities allowed to confirm the correlation between diet and functional fitness as well as the level of SWL, which is cognitive self-assessment of the quality of life related to health, holistically understood as the balance and integration of all dimensions constituting a human being (physiological, psychological and sociological).

The strength of the study seems to be including a broader spectrum of factors related to the behavioural determinants of health in a group of senior women who undertake organised physical activity to improve holistically defined health. The limitations of the presented work are primarily related to the nature of the group, including people interested in a healthy lifestyle, who voluntarily signed up for participation in the health activation programme for seniors, which limits the possibility of transferring the results to the general population of seniors. In subsequent research, changes in the analysed indicators (nutritional, somatic, fitness, and satisfaction with life) should also be taken into account at various stages of implementing the programme (at the beginning and after its completion). Simultaneously, the ambiguity of the obtained results indicates a larger spectrum of factors explaining the level of SWL, which may direct subsequent studies towards the development of a more complex model of explanatory factors.

## 5. Conclusions

Among the women who applied for participation in the “Healthy Active Senior” programme, an average level of satisfaction with life, national nutrition behaviours, low and average results in terms of functional fitness, and an increased amount of adipose tissue in body composition were demonstrated.An ambiguous relationship was found between life satisfaction and rational eating behaviours among women. Positive relationships concerned, e.g., consuming the recommended number of meals, wholegrain cereal products, avoiding alcoholic beverages, vitamin D supplementation, and proper fluid replenishment. On the other hand, the negative relationships mainly regarded the smaller scale of limiting atherogenic products (red meat, meat products, and animal fats).An ambiguous relationship was noted among women between life satisfaction and functional fitness indices, with positive ones related to endurance and physical performance as well as arm muscle strength, and negative ones to agility and dynamic balance.Among the women participating in the programme promoting health through physical activity, no significant relationships were exhibited between life satisfaction and somatic indices (BMI and basic body composition).The lack of full unambiguity concerning the results in the area of relationships between life satisfaction and nutritional behaviours as well as the functional fitness of senior women suggests the validity of further research, taking a wider spectrum of psycho-social, behavioural, and health variables into account.

## Figures and Tables

**Table 1 ijerph-20-01877-t001:** Level of rational nutrition behaviours among women participating in the “Healthy Active Senior” programme (N = 120) (descriptive statistics).

Nutrition Behaviours	Mean	Median	Min.	Max.	SD
Regular consumption of 5–6 meals a day	3.00	2.50	1.00	5.00	1.26
Regular consumption of 4–5 meals a day	3.32	4.00	1.00	5.00	1.14
Varied meals, freshly prepared	4.03	4.00	2.00	5.00	0.77
At least 1 hot meal a day	4.77	5.00	1.00	5.00	0.69
At least 2 l of fluids a day	3.49	4.00	1.00	5.00	1.22
Fruit and vegetables—3–5 times a day (approx. half of the nutritional ration)	3.38	4.00	1.00	5.00	1.18
Grain products, incl. whole-grains—daily	3.99	4.00	1.00	5.00	1.07
Dairy products, incl. those fermented—3 times a day	3.14	3.00	1.00	5.00	1.15
Incl. fish, eggs and lean meat in one’s diet	4.22	4.00	2.00	5.00	0.89
Sea fish 1–2 times a week	3.11	3.00	1.00	5.00	1.19
Limiting red and processed meats	4.07	4.00	1.00	5.00	1.04
Incl. legumes in one’s diet	3.49	4.00	2.00	5.00	1.05
Limiting animals fats	4.09	4.00	1.00	5.00	0.93
Plant-based oils—every day (or almost every day)	4.38	5.00	1.00	5.00	1.01
Limiting sugars, sweets, and sweetened beverages in one’s diet	4.22	4.00	1.00	5.00	1.01
Not adding salt to ready-made meals, choosing products with low table-salt content	4.20	4.00	2.00	5.00	0.86
Using spices (fresh and dried herbs)	4.24	5.00	2.00	5.00	0.94
Not consuming alcoholic beverages	3.90	4.00	1.00	5.00	1.23
Vitamin D 2000 IU supplementation—every day	4.07	5.00	1.00	5.00	1.36
Nutritional choices taking health state into account	4.32	5.00	1.00	5.00	0.89
Considering interactions between foods and medications	4.13	4.00	1.00	5.00	1.08
Physical activity—every day (or almost every day), at least 30–45 min	4.32	4.00	2.00	5.00	0.76
Index of national nutrition behaviours (total)	85.85	87.00	58.00	103.00	9.80

**Table 2 ijerph-20-01877-t002:** Level of body composition indices (BIA) and functional fitness indicators (Fullerton test) among women participating in the “Healthy Active Senior” programme (N = 120) (descriptive statistics).

Somatic Indices and Functional Fitness Indicators		Mean	Median	Min.	Max.	SD
BMI and body composition	BMI (kg/m^2^)	27.32	26.65	19.20	38.30	4.07
	Body mass (kg)	71.71	68.80	49.20	116.10	13.33
	FAT (%)	33.68	33.95	16.40	46.20	6.09
	FAT (kg)	24.26	23.10	2.70	40.90	7.61
	FFM (kg)	47.28	44.55	34.80	75.90	8.59
	TBW (kg)	34.61	32.60	25.50	55.60	6.28
Indices of functional fitness	‘Stand-Up from Chair’ within 30 s (repetitions)	15.37	15.00	10.00	22.00	2.53
	‘Flexing Loaded Arm’ within 30 s (repetitions)	18.36	18.00	13.00	27.00	2.81
	‘Seated Forward Bend’ (cm)	3.91	5.00	−27.00	28.50	14.66
	So-called ‘Safety Pin’ (Back Scratch) (cm)	−1.19	1.00	−20.00	16.50	8.16
	‘2.44-m Get-Up and Go’ (s)	4.83	4.67	3.30	6.66	0.69
	‘2-Minute Step in Place’ (steps)	112.62	111.00	91.00	158.00	12.32

**Table 3 ijerph-20-01877-t003:** Correlations between satisfaction with life (SWLS) and nutrition behaviours among women participating in the “Healthy Active Senior” programme (Spearman’s correlation analysis).

Nutrition Behaviours	Spearman’s R	t(N-2)	*p*-Value
Regular consumption of 5–6 meals a day	0.44	5.29	<0.001
Regular consumption of 4–5 meals a day	0.08	0.83	0.407
Varied meals, freshly prepared	0.05	0.55	0.582
At least 1 hot meal a day	−0.04	−0.45	0.655
At least 2 l of fluids a day	0.21	2.30	0.023
Fruit and vegetables—3–5 times a day (approx. half of the nutritional ration)	0.10	1.14	0.259
Grain products, incl. whole-grains—daily	0.29	3.34	0.001
Dairy products, incl. those fermented—3 times a day	−0.10	−1.04	0.302
Incl. fish, eggs, and lean meat in one’s diet	−0.08	−0.85	0.395
Sea fish 1–2 times a week	0.02	0.24	0.810
Limiting red and processed meats	−0.28	−3.22	0.002
Incl. legumes in one’s diet	−0.09	−1.03	0.307
Limiting animal fats	−0.18	−2.02	0.046
Plant-based oils—every day (or almost every day)	−0.21	−2.30	0.023
Limiting sugars, sweets and sweetened beverages in one’s diet	0.06	0.68	0.498
Not adding salt to ready-made meals, choosing products with low table-salt content	−0.05	−0.49	0.625
Using Spice (fresh and dried herbs)	0.03	0.33	0.739
Not consuming alcoholic beverages	0.20	2.19	0.030
Vitamin D 2000 supplementation—every day	0.23	2.61	0.010
Nutritional choices taking health state into account	−0.16	−1.81	0.072
Considering interactions between foods and medications	0.12	1.36	0.176
Physical activity—every day (or almost every day), at least 30–45 min	0.10	1.10	0.272
Index of national nutrition behaviours (total)	0.15	1.70	0.092

**Table 4 ijerph-20-01877-t004:** Correlations between satisfaction with life (SWLS) and somatic indices and indicators of functional fitness among women participating in the “Healthy Active Senior” programme (Spearman’s correlation analysis).

Variables	Spearman’s R	t(N-2)	*p*-Value
Body mass and SWLS	−0.00	−0.05	0.963
BMI and SWLS	−0.05	−0.55	0.581
FAT (%) and SWLS	−0.14	−1.56	0.120
FAT (kg) and SWLS	−0.14	−1.58	0.117
FFM (kg) and SWLS	0.15	1.66	0.100
TBW (kg) and SWLS	0.15	1.68	0.096
‘Stand-Up from Chair’ 30 s and SWLS	0.15	1.62	0.108
‘Flexing Loaded Arm’ 30 s and SWLS	0.18	1.98	0.050
‘Seated Forward Bend’ and SWLS	0.09	0.98	0.327
So-called ‘Safety Pin’ and SWLS	0.17	1.90	0.060
‘2.44-m Get-Up and Go’ and SWLS	−0.27	−3.01	0.003
‘2-Minute Step in Place’ and SWLS	0.20	2.15	0.034

**Table 5 ijerph-20-01877-t005:** Regression analysis—full model.

	Beta	Std. Err.—of Beta	B	Std. Err.—of B	t(109)	*p*-Value
Intercept			−7.81	11.73	−0.67	0.507
Age	0.31	0.09	0.35	0.10	3.57	0.001
‘Stand-Up from Chair’ 30 s	−0.15	0.11	−0.27	0.21	−1.30	0.196
‘Flexing Loaded Arm’	0.15	0.11	0.24	0.19	1.27	0.207
‘Seated Forward Bend’	0.09	0.10	0.03	0.03	0.89	0.376
So-called ‘Safety Pin’ (Back Scratch)	0.07	0.09	0.04	0.05	0.78	0.434
‘2.44-m Get-Up and Go’	−0.31	0.10	−2.14	0.71	−3.00	0.003
‘2-Minute Step in Place’	0.06	0.09	0.02	0.04	0.64	0.523
Rational nutrition behaviours (total index)	0.36	0.09	0.17	0.05	3.81	<0.001

R = 0.53; R2 = 0.28; Adjusted R2 = 0.23; F(8109) = 5.34; *p* < 0.001; Std. error of estimate: 4.13.

## Data Availability

Data available on request due to restrictions, e.g., privacy or ethical.

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
