# Peer review of "Satisfaction with Life and Nutritional Behaviour, Body Composition, and Functional Fitness of Women from the Kraków Population Participating in the “Healthy Active Senior” Programme"

_ijerph, 2023, doi:10.3390/ijerph20031877_

Round 1

Reviewer 1 Report

The article ‘Satisfaction with Life and Nutritional Behaviour, Body Composition and Functional Fitness of Women from the Kraków Population Participating in the "Healthy Active Senior" Programme’, presented for review is interesting. However, some issues require clarification. Below are some comments that may improve the article.

- If the program was aimed at seniors, why were people aged 54  already included in the study? What age are people considered seniors?

- Wasn't the increase in life satisfaction partly due to increased human contact when playing sports together with other people? Has this aspect been considered?

- In chapter 2.2. Instruments repeated text from Chapter 1. Introduction. The content does not match the title of the chapter.

- Why, when describing the results, factors that had no significant impact were omitted. According to the reviewer, it is also worth highlighting them.

- Conclusions section needs redrafting.

Reviewer 2 Report

   The main objective of this study is to determine the relationship between satisfaction with life (SWL), nutritional behaviours, somatic indices and functional efficiency of senior women. The topic is interesting, however, need some improvement.

1.       Based on what familiar definition the authors considered age between 54-84 as senior women?

2.       How did authors measure sample size? How 120 participants are enough for this study?

3.       Sampling process is not clear. Need more explanations how they reach to the participants and etc.

4.       2020 is pandemic time. Why didn’t the authors consider pandemic variables in their research model?

5.       There are some studies in 2021 and 2022 which are no considered by the authors.

6.       Please prepare reliability and validity of your questionnaire.

7.       I would like to see about their analysis regarding their missing data and outliers.

8.       I would like to see the pilot test outputs.

9.       What is the novelty of the study?

Round 2

Reviewer 1 Report

The comments indicated in the review have been corrected by the authors to a satisfactory extent.

Reviewer 2 Report

The authors amended all of my comments. Thank you!